# MSFF-Net: Multi-Sensor Frequency-Domain Feature Fusion Network with Lightweight 1D CNN for Bearing Fault Diagnosis

**DOI:** 10.3390/s25144348

**Published:** 2025-07-11

**Authors:** Miao Dai, Hangyeol Jo, Moonsuk Kim, Sang-Woo Ban

**Affiliations:** 1Department of Information & Communication Engineering, Graduate School, Dongguk University, Gyeongju 38066, Republic of Korea; daimiao@dongguk.ac.kr (M.D.); johangyeol0309@gmail.com (H.J.); anstjr555@gmail.com (M.K.); 2Department of Electronics, Information & Communication Engineering, Dongguk University, Gyeongju 38066, Republic of Korea

**Keywords:** bearing fault diagnosis, one-dimensional convolutional neural network, acoustic data, Vibration data, Multi-sensor data fusion

## Abstract

**Highlights:**

**What are the main findings?**
A lightweight frequency-domain fusion model is proposed to integrate vibration and acoustic signals for bearing fault diagnosis.

**What is the implication of the main finding?**
The proposed model achieves high diagnostic accuracy, even under limited-data conditions.The approach facilitates efficient multi-sensor integration through compact architecture, without relying on elaborate preprocessing.

**Abstract:**

This study proposes MSFF-Net, a lightweight deep learning framework for bearing fault diagnosis based on frequency-domain multi-sensor fusion. The vibration and acoustic signals are initially converted into the frequency domain using the fast Fourier transform (FFT), enabling the extraction of temporally invariant spectral features. These features are processed by a compact one-dimensional convolutional neural network, where modality-specific representations are fused at the feature level to capture complementary fault-related information. The proposed method demonstrates robust and superior performance under both full and scarce data conditions, as verified through experiments on a publicly available dataset. Experimental results on a publicly available dataset indicate that the proposed model attains an average accuracy of 99.73%, outperforming state-of-the-art (SOTA) methods in both accuracy and stability. With only about 70.3% of the parameters of the SOTA model, it offers faster inference and reduced computational cost. Ablation studies confirm that multi-sensor fusion improves all classification metrics over single-sensor setups. Under few-shot conditions with 20 samples per class, the model retains 94.69% accuracy, highlighting its strong generalization in data-limited scenarios. The results validate the effectiveness, computational efficiency, and practical applicability of the model for deployment in data-constrained industrial environments.

## 1. Introduction

Bearings constitute critical components in rotating machinery and are susceptible to failure under harsh operating conditions. Undetected faults may propagate, potentially leading to system breakdowns, safety risks, and substantial economic losses [1]. Therefore, accurate bearing fault diagnosis is essential for ensuring operational reliability.

Bearing fault diagnosis is conventionally conducted through sensor-based signal analysis [2]. With the rapid advancement of deep learning, data-driven approaches have attracted considerable attention due to their capability to autonomously extract discriminative features from raw sensor data, thereby reducing the dependency on manual feature engineering. Recent approaches—such as transformers [3,4], convolutional neural networks (CNN) [5,6], attention mechanisms (AM) [7], and hybrid architectures [8,9]—have demonstrated competitive performance under diverse operational conditions. However, most of these methods primarily rely on single-sensor inputs. While effective under controlled laboratory settings, single-sensor approaches often encounter limitations in real-world industrial environments due to factors such as sensor placement constraints, environmental noise, and restricted spatial coverage, which may result in fault misclassification or missed detections [10].

In contrast, multi-sensor fusion leverages complementary information from diverse sensing modalities, thereby enhancing diagnostic accuracy and robustness [11]. For example, acoustic signals enable non-contact measurement and are effective for early fault detection [12,13], whereas vibration sensors exhibit greater sensitivity to impact-related faults and offer a cost-effective means for continuous condition monitoring [14,15]. The integration of these modalities further improves the accuracy and reliability of fault characterization in real-world monitoring scenarios. Therefore, combining multiple sensors provides an effective means of further enhancing the accuracy and reliability of bearing fault diagnosis.

Multi-sensor data fusion techniques are generally categorized into three levels: data-level fusion, feature-level fusion, and decision-level fusion [16]. Data-level fusion involves the integration of raw signals from multiple sensors at the input stage with the goal of preserving the maximum amount of original information [17,18,19]. For instance, Gong et al. [20] proposed a method that converts multi-sensor signals into two-dimensional time–frequency representations, which are subsequently analyzed using a hierarchical vision transformer. This approach demonstrated strong performance across diverse datasets, exhibiting notable resilience to noise and commendable generalization capacity under limited-data conditions. However, it was determined that input redundancy was a limiting factor affecting scalability. In a similar manner, S. Wang et al. [21] utilized variational mode decomposition to transform signals into gray-scale images, which were subsequently processed using an enhanced Google-Net architecture. Despite the fact that the method demonstrated enhanced robustness, it was nevertheless asserted that its dependence on manually specified preprocessing parameters, in addition to its limited capacity for generalization in small-sample scenarios, were significant drawbacks. In another study, Y. Dong et al. [22] proposed a lightweight method that employed a metric-based cliff entropy preprocessing technique prior to the input of data into the Diwave-former model. While this approach did indeed achieve computational efficiency, its reliance on vibration-domain analysis meant that its applicability to other sensor modalities was constrained. When considered as a whole, these data-level fusion studies underscore shared challenges, including input redundancy and the difficulty of effectively integrating heterogeneous sensor data, which can impede practical deployment in real-world applications. Feature-level fusion is the process of independently extracting features from each sensor and subsequently integrating them within a unified representation. This enables more flexible handling of heterogeneous multi-sensor data [23,24,25,26]. A CNN-based feature fusion method designed with a multi-task learning module was proposed by J. Cui et al. [27], which effectively achieved high-accuracy fault diagnosis across multiple sensors. However, the requirement to manually specify certain parameters involved in the multi-task learning process remains a limitation. A methodology combining short-time Fourier transform (STFT) and CNN was introduced by Gültekin et al. [28], through which promising performance was demonstrated on edge devices. However, performance degradation was observed under complex data conditions. The signal-to-image encoding unit (SIEU), incorporating the improved constant-Q non-stationary Gabor transform (ICQ-NSGT), was employed, and the coarse-to-fine dual-scale time–frequency attention fusion network (CDTFAFN) was proposed by X. Yan et al. [29]. While strong noise robustness was achieved, the method relied on handcrafted signal transformations and involved a complex post-fusion processing stage, resulting in a relatively large parameter count. Despite recent advancements in the field, existing studies suggest that feature-level fusion methods continue to encounter substantial challenges, particularly with regard to architectural complexity and the necessity of domain expertise for the alignment and integration of heterogeneous features. Decision-level fusion is a process that involves the aggregation of outputs from multiple models with a view to enhancing the accuracy and robustness of the resulting decision [30,31]. A global–local temporal encoder combined with a cross-sensor correlative channel-aware fusion strategy was proposed by Z. Feng et al. [32]. This strategy was shown to result in high diagnostic accuracy. Nonetheless, its applicability was restricted for two principal reasons. Firstly, there was insufficient generalization in small-sample training scenarios. Secondly, there was a need for manual parameter tuning. A fusion method integrating multi-scale CNN with fuzzy rank-based techniques was introduced by J. Tong et al. [33], which exhibited superior performance in comparison to basic baseline models. However, the final fusion process was relatively static and highly dependent on sensor-specific characteristics, thereby requiring reconfiguration when applied to other sensor types. It is evident that, in general, decision-level fusion approaches frequently depend on fixed aggregation schemes. Such schemes are characterized by their inability to adapt and their lack of robustness. Moreover, these schemes may introduce implicit statistical assumptions about sensor inputs. Furthermore, these methodologies have the capacity to result in an augmentation of model intricacy and the prospective forfeiture of intermediate feature representations during the culminating decision-making phase.

In summary, contemporary multi-sensor fusion strategies frequently encounter several practical limitations. These include high information redundancy, strong dependence on domain-specific preprocessing, complex fusion architectures, insufficient generalization under small-sample conditions, and large parameter footprints. Consequently, their applicability to real-world diagnostic scenarios remains limited.

In order to address the limitations commonly observed in existing feature-level fusion approaches—such as excessive model complexity, strong reliance on handcrafted processing, and limited robustness under data-scarce conditions—this study proposes a frequency-domain feature-level fusion framework, termed MSFF-Net (multi-sensor frequency-domain feature-level fusion network), for bearing fault diagnosis. MSFF-Net employs a streamlined and well-structured design to model frequency-domain information from multiple sensors. Deep learning techniques are utilized to extract and integrate core feature representations from each sensor input. This enables effective fault identification without the necessity of elaborate preprocessing or excessively complex model components.

The principal contributions of this study are as follows. The proposal of a frequency-domain feature-level fusion framework, termed MSFF-Net, is presented with the objective of facilitating effective bearing fault diagnosis. In this framework, raw signals from multiple sensors are subjected to the Fourier transform, and essential features are extracted using parallel 1D convolutional structures. The extracted features are subsequently fused via feature-level concatenation. The experimental results demonstrate that MSFF-Net achieves a consistently favorable diagnostic performance in comparison with a range of representative fusion-based approaches, particularly under conditions of data scarcity. This finding highlights the potential for the practical deployment of MSFF-Net. Moreover, the network employs a structurally concise and computationally efficient design, with the objective of reducing reliance on complex preprocessing and handcrafted fusion mechanisms, while maintaining the capability to effectively model and integrate heterogeneous sensor information.

The remainder of this paper is organized as follows. Section 2 details the proposed MSFF-Net framework for multi-sensor bearing fault diagnosis. Section 3 describes the experimental setup, evaluation metrics, and presents a comprehensive analysis of the results. Section 4 concludes the paper by summarizing key findings and discussing limitations as well as potential directions for future research.

## 2. Theory and Methodology

### 2.1. Fast Fourier Transform

In the context of rotating machinery, the presence of mechanical faults has been observed to give rise to the manifestation of localized spectral patterns at specific characteristic frequencies [34]. As demonstrated in previous research [12], it has been established that bearing degradation can result in alterations to the frequency band and shifts, particularly in the context of acoustic signals. It is evident that these observations have provided the impetus for the present study, which has been designed to apply the fast Fourier transform (FFT) to both vibration and acoustic signals with a view to capturing relevant spectral features. FFT is a transformative technique that transforms time-domain signals into frequency-domain representations, thereby accentuating fault-related components and enabling more robust and informative feature extraction [35]. Moreover, the FFT is a computationally efficient algorithm that reduces the complexity of the Discrete Fourier Transform (DFT), rendering it suitable for real-time diagnostic applications. The formula for DFT is shown in Equation (1):(1)$$X\left[k\right]=\sum_{n=0}^{N-1}x\left[n\right]e^{-j\;2\pi \frac{nk}{N}},\;\;k=0,1,2,\ldots ,N-1$$
where X[k] represents the frequency spectrum and x[n] denotes the original time-domain signal. The magnitude spectrum obtained from the FFT, retained in accordance with the Nyquist sampling theorem, is utilized as the input data for the proposed bearing fault diagnosis model.

### 2.2. Z-Score Normalization

Due to substantial variations in the magnitude of frequency components following FFT, normalization is applied to the data to facilitate stable and efficient model training. In this study, Z-score normalization is adopted, as defined in Equation (2):(2)$$\begin{array}{c} x_{k}^{n}=\frac{X_{k}^{n}-\mu }{\sigma },\;\;n=1,\;2,\ldots ,M;\;\;k=0,\;1,\ldots ,\;N-1 \end{array}$$
where $x_{k}^{n}$ is the standardized signal at the k-th sample point of the n-th sample (with M being the total number of samples), and $X_{k}^{n}$ is the corresponding raw input value. The terms μ and σ are scalar values representing the global mean and standard deviation computed across all samples and all sample points in the dataset.

Z-score normalization is applied to mitigate the influence of differing magnitudes across frequency components, which could otherwise hinder model convergence. In this preprocessing step, the input scale is standardized to improve training stability and promote balanced feature contributions. Additionally, values are clipped to the range [−3,3] to suppress the influence of outliers and noise in the frequency domain.

### 2.3. One-Dimensional Convolutional Neural Network

One-dimensional convolutional neural networks (1D CNN) are widely adopted due to their inherent suitability for modeling sequential data structures [36], as well as their significantly reduced computational complexity relative to two-dimensional CNN architectures [37]. The convolution operation is conducted by sliding a learnable kernel across the input sequence to capture local contextual features. The resulting convolutional response $z_{i}$ at position i for the n-th input samples is formally expressed in Equation (3):(3)$$z_{i}=\sum_{m=0}^{K-1}x_{i+m}\cdot w_{m}+b$$
where $x_{i+m}$ denotes the input value at position i+m of the n-th sample, $w_{m}$ represents the kernel weight at position m, b is the bias term, and K denotes the kernel size.

In each convolutional block, a Leaky Rectified Linear Unit (Leaky ReLU) activation function is applied to introduce nonlinearity into the feature maps. Leaky ReLU allows small gradients to propagate for negative input values, thereby improving gradient flow during backpropagation. The activated output $a_{i}$ is computed as defined in Equation (4):(4)$$a_{i}=LeakyReLU\left(z_{i}\right)=\left\{\begin{array}{c} z_{i},\;\;if\;z_{i}\geq 0\\ \alpha z_{i},\;if\;z_{i}<0 \end{array}\right.$$
where α is a small constant typically set to 0.01.

One-dimensional max pooling is applied to reduce dimensionality while retaining salient features. The pooled output $p_{i,c}$ for channel c at index i is computed as defined in Equation (5):(5)$$\begin{array}{c} p_{i,c}=max\left(a_{j,c}|j\in \left\{s\cdot i,\ldots ,s\cdot i+k-1\right\}\right) \end{array}$$
where $a_{j,c}$ is the activated value at position j in channel c, k is the pooling window size, and s is the stride. Non-overlapping pooling is implemented by setting the stride equal to the window size.

Feature-level concatenation is employed to alleviate the potential feature entanglement that may occur with other fusion methods. The final fused feature vector p is obtained by concatenating the flattened features $p_{s1}$ and $p_{s2}$ from the two sensors along the feature dimension, as shown in Equation (6):(6)$$p={\;p}_{s1}⨁p_{s2}$$

The final fused feature vector p is passed through a fully connected (FC) layer. The linear transformation output f of the FC layer is computed as defined in Equation (7):(7)$$f=W_{fc}\cdot p+b_{fc}$$
where $W_{fc}$ is the weight matrix and $b_{fc}$ is the bias vector.

A ReLU (rectified linear unit) activation function is then applied, as shown in Equation (8):(8)$$a=ReLU\left(f\right)=\left\{\begin{array}{c} f,\;\;if\;f\geq 0\\ 0,\;\;if\;f<0 \end{array}\right.$$

ReLU is employed in the fully connected layers to suppress weak activations and maintain nonlinearity.

In multi-class classification tasks, a linear transformation is applied in the final layer, followed by the softmax activation function, which converts the un-normalized outputs into a probability distribution over C classes, as defined in Equation (9):(9)$${\hat{y}}_{k}=\frac{\exp\left(f_{k}\right)}{\sum_{j=1}^{C}\exp\left(f_{j}\right)}$$
where $f_{k}$ is the score for class k, and ${\hat{y}}_{k}$ is the predicted probability for class k and C is the total number of classes.

### 2.4. MSFF-Net: A Multi-Sensor Frequency-Domain Feature-Level Fusion Model

MSFF-Net consists of three core modules: feature extraction, feature fusion, and classification. During the feature extraction phase, sensor data from different modalities are individually processed using a 1D CNN to extract deep hierarchical representations that capture localized patterns and fault signatures. In the feature fusion phase, features from multiple sensors are integrated via feature-level concatenation, allowing the model to exploit complementary diagnostic information across modalities and thereby improve the robustness and discriminability of the learned representations. Finally, in the classification phase, the fused features are passed through fully connected layers to perform multi-class fault identification, enhancing the model’s generalization capability.

To improve training stability and reduce overfitting, batch normalization (BN) and dropout are incorporated into MSFF-Net. BN is applied before activation functions in convolutional layers to normalize intermediate feature distributions, thereby reducing internal covariate shift and facilitating convergence. Dropout is employed during training to randomly deactivate neurons, introducing regularization that prevents overfitting and improves feature robustness.

In the feature extraction module, a series of 1D convolutional layers is utilized to progressively extract informative features from the input data. Max pooling layers are employed to retain salient representations while reducing feature resolution and computational cost. At later stages, 1 × 1 convolutions are introduced to reorganize channel-wise information. To further enhance regularization, dropout with a rate of 0.25 is applied between convolutional blocks, helping to suppress overfitting. The detailed architecture of the feature extraction module is illustrated in Figure 1.

In the subsequent feature fusion phase, features extracted from multiple sensors are integrated via feature-level concatenation. To mitigate the computational inefficiencies associated with high-dimensional tensor operations, the features are first flattened into one-dimensional vectors prior to fusion. The resulting concatenated feature vector is then passed through fully connected layers to further refine the learned representations and facilitate high-level feature interactions.

To improve generalization and mitigate overfitting, dropout with a rate of 0.5 is applied between the fully connected layers as a regularization strategy, effectively reducing co-adaptation among neurons and enhancing model robustness. The overall architecture of MSFF-Net, including the feature extraction and fusion stages, is illustrated in Figure 2.

The model is trained under a supervised learning framework, employing categorical cross-entropy as the loss function for multi-class classification tasks. Under this framework, the divergence between the predicted probability distribution and the true class labels is measured by the cross-entropy loss, and the resulting gradients are used to update model parameters through backpropagation.

## 3. Experiments

### 3.1. Experimental Setup

The effectiveness of the proposed methodology is validated through experiments conducted using the University of Ottawa Rolling-Element Dataset—Vibration and Acoustic Faults under Constant Load and Speed conditions (UORED-VAFCLS) [38]. As illustrated in Figure 3, a PCB 623C01 accelerometer (PCB Piezotronics, Inc., Depew, NY, USA) is mounted on the drive-end bearing, while a PCB 130F20 microphone (PCB Piezotronics, Inc., Depew, NY, USA) is positioned approximately 2 cm from the bearing surface. Vibration and acoustic signals are synchronously acquired using a National Instruments USB-6212 data acquisition system at a sampling rate of 42,000 Hz, with each sample lasting 10 s.

The dataset includes measurements from degreased NSK 6203ZZ (NSK Ltd., Shinagawa-ku, Tokyo, Japa) and FAFNIR 203KD (The Timken Company, North Canton, OH, USA) bearing models. To ensure consistency and prevent cross-contamination, a new bearing is installed prior to each test. Nine bearing states are evaluated, covering four fault types—inner race, outer race, ball, and cage—across three degradation stages: healthy, developing fault, and faulty. All tests are conducted under a rotational speed of 1750 rpm and a 400 N radial load, except for ball faults, which are acquired under no-load conditions. Only data collected from FAFNIR 203KD bearings are utilized for model training and evaluation.

For each class, 50 vibration samples and 50 acoustic samples are randomly selected for training, with an equal number allocated for testing to ensure class balance. To facilitate fault identification, a systematic label encoding scheme is adopted, where fault locations are represented by uppercase initials (‘I’ for inner race, ‘O’ for outer race, ‘B’ for ball, and ‘C’ for cage), and fault stages are denoted by ‘H’ (healthy), ‘P’ (in progression, representing developing faults), and ‘C’ (complete, representing fully developed faults). The data partition and corresponding label definitions are described in Table 1.

Raw time-domain signals are segmented into fixed-length windows of 2048 samples. According to Yoo et al. [5], using one mechanical revolution as input allows for dimensionality reduction in lightweight CNN while mitigating accuracy loss. In the UORED-VAFCLS dataset, the nominal speed is 1750 rpm, and the sampling rate is 42,000 Hz, so one revolution corresponds to approximately 1440 samples. Considering practical speed fluctuations, the raw signal is segmented into windows of 2048 samples (a power of two) to enhance FFT computation and model input efficiency. Each window is then transformed into a 1024-point frequency-domain vector via the FFT, emphasizing spectral characteristics while discarding temporal information. Subsequently, Z-score normalization is applied to standardize the transformed data and enhance feature stability across different samples. Representative FFT power spectra derived from vibration and acoustic signals are illustrated in Figure 4. The elimination of temporal variations in the frequency domain enables clearer identification of fault-related spectral components.

The proposed MSFF-Net is trained using the Adam optimizer with a learning rate of 0.001, a batch size of 16, and 100 epochs. Implementation is carried out in Python 3.8.0 and TensorFlow 2.10.0, and the model is executed on a workstation equipped with an Intel i7–11700KF CPU, 128 GB RAM, and an NVIDIA RTX 3080 Ti GPU.

### 3.2. Performance of the Proposed Method

The performance stability of the proposed MSFF-Net was evaluated over 10 independent trials, with classification results recorded for each run. Given the complexity of the task, a comprehensive assessment was carried out based on five standard metrics: accuracy, precision, recall, specificity, and F1-score. In bearing fault diagnosis, both false alarms and missed detections can lead to serious operational and safety issues. Accordingly, precision, recall, specificity, and F1-score were employed to quantitatively evaluate the model’s diagnostic performance across key classification criteria. Precision measures the proportion of correctly identified fault cases among predicted faults and is essential for avoiding unnecessary maintenance. Recall, or sensitivity, quantifies the model’s ability to detect actual faults, ensuring early degradations are not missed. Specificity indicates the proportion of correctly classified healthy samples, thereby reducing false positives. The F1-score, the harmonic mean of precision and recall, offers a balanced evaluation under class imbalance or limited data.

As shown in Table 2, the proposed model achieved consistently high performance across 10 independent trials. The average test accuracy reached 99.73%, with average values of precision, recall, and specificity all exceeding 99%. The F1-score likewise attained 99.73%. In addition, the standard deviations for all metrics were minimal, confirming the model’s stability and robustness.

As shown in Table 3, the proposed vibration-acoustic fusion (VAF) configuration exhibits superior performance across all evaluation metrics compared to single-modality configurations. This result provides empirical evidence supporting the diagnostic effectiveness of multi-sensor fusion in bearing fault classification. The VAF configuration consistently achieves high precision and balanced classification outcomes, with an average accuracy of 99.73% and an F1-score of 99.73%.

When compared to the vibration-only configuration (VIB), which utilizes solely vibration signals, VAF demonstrates a performance improvement of 1.33%p in accuracy and 1.37%p in the F1-score. Although the improvement appears moderate in magnitude, it is consistently observed across repeated trials, suggesting that the inclusion of acoustic signals contributes discriminative features that are not fully captured by vibration signals alone, particularly under complex or subtle fault conditions.

In contrast, the acoustic-only configuration (ACO), which utilizes solely acoustic signals, exhibits substantially lower performance, achieving an average accuracy of 85.91% and an F1-score of 84.35%. These values correspond to a 13.82%p and 15.38%p decrease, respectively, when compared to the VAF configuration. Furthermore, with respect to specificity, the ACO configuration records 98.24%, which is 1.56%p lower than the VIB configuration. These results indicate that the likelihood of false positive classifications increases when relying exclusively on acoustic data. This degradation in performance may be attributed to the inherently lower signal-to-noise ratio of acoustic signals and the limited discriminability of certain defect signatures when acoustic information is used in isolation.

To further analyze the feature representations learned by different model configurations, we visualize them in a reduced-dimensional space. Although t-SNE [39] is commonly used for visualizing feature separability, it introduces randomness and is sensitive to parameter choices, which can lead to inconsistent results across runs. To compare the feature representations across different modality-based models in a stable and consistent manner, Linear Discriminant Analysis (LDA) [40] is employed. LDA aims to find a projection matrix W that maximizes the ratio of between-class to within-class scatter, formulated as Equation (10):(10)$$W^{*}=\underset{W}{argmax}\frac{\left|W^{T}S_{B}W\right|}{\left|W^{T}S_{W}W\right|}$$
where $S_{B}$ and $S_{W}$ are defined as Equations (11) and (12):(11)$$S_{B}=\sum_{i=1}^{C}N_{i}\left(\mu_{i}-\mu \right){\left(\mu_{i}-\mu \right)}^{T}$$(12)$$S_{w}=\sum_{i=1}^{C}\sum_{x\in D_{i}}\left(x-\mu_{i}\right){\left(x-\mu_{i}\right)}^{T}$$
where C is the number of classes, $N_{i}$ is the number of samples in class, $\mu_{i}$ is the mean vector of class, μ is the overall mean vector, and $D_{i}$ is the set of samples in class i. This formulation allows LDA to enhance class separability by finding projections that compact intra-class distributions while maximizing inter-class distances.

To ensure consistency and mitigate the effects of random sampling, a single trained model instance was used for analysis. For each of the 10 independent trials, 200 samples were randomly selected per class. The evaluation was conducted at the final representation stage, referring to the feature vectors immediately preceding the classification layer.

In this context, LDA was applied not merely as a visualization tool, but as a method for quantitatively evaluating the degree of class separability in the learned feature space. Specifically, LDA was used to project the high-dimensional feature vectors into a lower-dimensional space, enabling the computation of three discriminability metrics: intra-class distance, inter-class distance, and their ratio (inter/intra). These metrics collectively provide a numerical assessment of the structural discriminability achieved by each model configuration. The reported values represent the mean results over the 10 trials.

Figure 5 shows the LDA visualization results for different model configurations. To enable a fair comparison, the same coordinate scale is applied to all modality-specific plots. Among the three configurations, the VAF model exhibits the most distinct and well-separated class boundaries, indicating that its fused representations result in higher class discriminability. In contrast, the VIB and ACO models display varying degrees of class overlap, particularly in fault categories with subtle spectral differences, suggesting that single-modality configurations offer limited separability in the feature space.

A similar trend is observed in the quantitative discriminability metrics presented in Table 4, which summarizes the results across different modality-based input configurations. While the intra-class distance of the VAF configuration remains comparable to those of the other methods, it achieves the highest inter-class distance among the three. As a result, the inter/intra ratio for VAF is the most pronounced, indicating superior structural separability in the learned feature space. This suggests that the fused representations obtained through multi-sensor integration enhance the model’s ability to distinguish between fault classes more effectively than single-modality approaches.

### 3.3. Comparison with Other Methods

To evaluate the effectiveness of the proposed MSFF-Net, comparative experiments were conducted using the same dataset and evaluation protocols as those employed in the main experiments. All baseline models were trained under identical conditions, including the data split ratio and key hyperparameters such as batch size, learning rate, number of epochs, and optimizer settings. A uniform input size was also applied across all models to ensure consistency in data representation, thereby facilitating a fair and meaningful comparison.

For a comprehensive assessment, three baseline methods were selected, each representing a distinct approach to sensor signal modeling and fusion. The first comparative model, referred to as 1D-CNN-based VAF [10], utilizes a one-dimensional convolutional neural network to directly process raw vibration and acoustic signals. Its architecture features large convolutional kernels with a gradually decreasing number of filters across layers. Feature extraction is carried out through a sequence of convolutional operations, and the extracted features from both modalities are concatenated and forwarded to fully connected layers for classification. This model has previously demonstrated effective performance on a self-collected dataset, highlighting its practical applicability in dual-modality signal analysis.

The second comparative model is CDTFAFN [29], a time–frequency fusion network developed for heterogeneous sensor inputs. It transforms raw signals into time–frequency representations using an enhanced constant-Q non-stationary Gabor transform, followed by dual-scale attention-based fusion and fully connected classification. CDTFAFN has shown strong performance under noisy conditions and achieved state-of-the-art results on the University of Ottawa Rolling-Element Dataset.

The third method, MSCNN [33], employs a decision-level fusion strategy based on a multi-scale convolutional neural network. Sensor signals are processed in parallel through multiple branches, and final decisions are aggregated using a fuzzy rank-based fusion scheme. The model has demonstrated reliable diagnostic performance across two public bearing fault datasets.

The selected baseline models were chosen to represent diverse architectural paradigms and fusion strategies. A conventional single-stage convolutional approach is reflected in the 1D-CNN-based VAF model, which directly processes raw multi-sensor inputs. CDTFAFN represents a more complex architecture that integrates handcrafted time–frequency transformation with multi-scale attention-based feature fusion. In contrast, MSCNN adopts a decision-level fusion scheme, where sensor-specific branches operate in parallel and final predictions are obtained via score-level aggregation. Collectively, these models provide a representative basis for evaluating the diagnostic effectiveness, computational efficiency, and generalization capability of the proposed MSFF-Net. For CDTFAFN, we followed the same dataset partitioning scheme as reported in the original study [29]. To ensure a fair comparison, the performance results were directly cited from the published work without modification.

The quantitative evaluation results are presented in Table 5, which summarizes four key performance metrics. These are: (1) the average classification accuracy, representing the overall diagnostic capability of each model; (2) the standard deviation of accuracy, indicating the stability and consistency of performance across multiple runs; (3) the elapsed time, measuring the average total computational cost encompassing both training and inference phases; and (4) the number of trainable parameters, which serves as an indicator of model complexity and its suitability for deployment in resource-constrained environments.

As shown in Table 5, the proposed MSFF-Net achieves the highest average classification accuracy of 99.73%, along with the lowest standard deviation of 0.0025, indicating strong classification performance and high stability across trials. Compared with the 1D-CNN-based VAF approach, which achieves an accuracy of 97.27%, MSFF-Net shows an improvement of approximately 2.46%p while maintaining a compact model structure. The consistently low variance also suggests greater reliability across repeated evaluations.

With respect to computational efficiency and model complexity, MSFF-Net exhibits a well-balanced performance. Although it does not achieve the fastest processing time—averaging 49.69 s—it is notably more efficient than CDTFAFN and MSCNN, which require 71.03 and 70.02 s, respectively. The 1D-CNN-based VAF model attains the lowest processing time (30.07 s) and the smallest parameter count (1.7 million); however, this comes at the expense of diagnostic accuracy, which is 2.46 percentage points lower than that of MSFF-Net. In contrast, MSFF-Net achieves superior classification performance while maintaining a moderate model size of 2.4 million parameters, highlighting its practical advantage for scenarios requiring a balance between accuracy and deployability.

Compared to CDTFAFN, which incorporates complex signal transformations and multi-scale attention-based fusion, MSFF-Net achieves a marginal improvement in classification accuracy with a gain of 0.2%p. Furthermore, it reduces the parameter count by 29.4%, indicating enhanced computational efficiency and a more lightweight model architecture.

In the case of MSCNN, the average classification accuracy is 87.38%, which is 12.35%p lower than that of MSFF-Net, despite having a relatively small model size of 0.3 million parameters.

In summary, the experimental findings demonstrate that MSFF-Net achieves a well-balanced trade-off among classification accuracy, performance stability, computational time, and model complexity. Such characteristics are particularly advantageous in practical fault diagnosis applications, where timely and reliable results must be obtained under limited computational and time resources.

### 3.4. Small-Sample Experiment

In real-world industrial applications, the availability of labeled fault data is often limited due to factors such as equipment reliability, operational cost, and safety constraints. Under these conditions, the generalization capability of diagnostic models becomes a critical factor. To evaluate model performance under limited-data scenarios, a small-sample experiment was conducted by progressively reducing the number of training samples per class to 50, 40, 30, and 20, while maintaining a fixed test set configuration. Each experimental setting was repeated over 10 independent trials, and the mean classification accuracy was computed to evaluate performance.

Although the dataset employed in this study is inherently class-balanced by design, the small-sample experiments may still introduce implicit data imbalance during training due to stochastic sampling effects inherent in gradient-based optimization. This form of imbalance, which does not stem from the underlying data distribution, can nevertheless result in underfitting or unintended shifts in the decision boundary, particularly affecting minority-class representations during the early stages of model optimization.

To quantitatively assess the model’s discriminative capability under limited-data conditions, the area under the receiver operating characteristic curve (AUC) was employed. This metric evaluates the trade-off between true positive and false positive rates across different classification thresholds.

Model evaluation was performed using both micro-average AUC and macro-average AUC metrics. The micro-average AUC is computed by aggregating predictions across all samples, reflecting overall classification performance. In contrast, the macro-average AUC is obtained by averaging the AUC scores for each class individually, assigning equal weight regardless of class frequency. This twofold evaluation provides a more comprehensive view of model behavior, particularly under small-sample settings where class-level performance differences may otherwise be masked. The AUC curves of MSFF-Net based on these two averaging strategies are illustrated in Figure 6.

As shown in Figure 6, MSFF-Net consistently maintains high classification reliability across varying training sample sizes, as evidenced by both micro- and macro-averaged ROC curves. Although slight reductions in AUC are observed at 30 and 20 samples per class, all values remain above 0.9990, indicating negligible performance degradation. The near-identical shapes of the two curves further suggest that both global recognition capability and class-level consistency are well preserved. These findings suggest that even under limited-data conditions, the model remains robust under limited-data conditions and is resilient to potential class imbalance effects, without exhibiting significant deterioration in class-specific performance.

To further validate this robustness, the baseline methods introduced in Section 3.3 were re-assessed under the same reduced-sample protocol, wherein the number of training instances per class was progressively decreased. The comparative results under these constrained settings are presented in Figure 7. For CDTFAFN, the results were reproduced based on the original experimental setup and data splits reported in [29]. Detailed classification accuracies for all models across different training sample sizes are summarized in Table 6, with the CDTFAFN results directly cited from [29].

As shown by the trend in Figure 7 and the results summarized in Table 6, all models experience a decline in classification accuracy as the number of training samples per class decreases. Despite this general performance degradation, the proposed MSFF-Net consistently attains the highest accuracy across all sample sizes, evidencing its robust generalization capability under limited-data conditions. Notably, in the most constrained scenario with only 20 training samples per class, MSFF-Net achieves an accuracy of 94.69%, surpassing CDTFAFN (92.33%), MSCNN (82.89%), and the 1D-CNN-based VAF model (89.15%).

Compared to the other 1D CNN-based model, MSFF-Net not only achieves superior performance under full-data conditions but also exhibits greater robustness as the amount of training data decreases. Between 50 and 20 samples per class, MSFF-Net’s classification accuracy declines by 5.04%p, whereas the 1D-CNN-based VAF model shows a more pronounced drop of 8.12%p. Although CDTFAFN performs competitively when ample training data is available, its accuracy decreases by 7.20%p under data-scarce conditions, which may be attributed to its dependence on complex time–frequency transformations and multi-scale fusion mechanisms. While MSCNN maintains relatively stable performance at higher sample sizes, its accuracy drops by 4.49%p between 30 and 20 samples per class, resulting in an 11.8%p performance gap compared to MSFF-Net in the most limited-data setting.

These results indicate that MSFF-Net achieves a favorable balance between classification accuracy and performance stability, even under limited-data conditions. This robustness highlights its potential for practical deployment in real-world fault diagnosis applications, where the availability of labeled data is often restricted.

### 3.5. Analysis and Discussion of Experimental Results

The experimental results demonstrate that the proposed MSFF-Net consistently achieves high diagnostic performance across various evaluation settings. As presented in Table 2, the model exhibits low standard deviations across 10 independent trials and achieves high values in all five classification metrics—accuracy, precision, recall, specificity, and F1-score. These outcomes indicate the model’s stability and reliability in performing multi-class fault diagnosis tasks.

Table 3 presents a comparative analysis between the full fusion-based configuration of MSFF-Net (VAF) and its single-modality variants. The fused model consistently outperforms both the vibration-only (VIB) and acoustic-only (ACO) configurations across all evaluation metrics, thereby confirming that the integration of vibration and acoustic signals enhances classification performance. This advantage is further substantiated by the results shown in Figure 5 and Table 4, where the fused configuration demonstrates improved feature separability. These findings suggest that effective multi-sensor integration at the feature level can significantly improve diagnostic accuracy, even in the absence of complex fusion mechanisms.

Under limited-data conditions, MSFF-Net continues to demonstrate robust performance. As presented in Figure 7 and Table 6, the model consistently achieves the highest accuracy across all training sizes—from 50 to 20 samples per class—while exhibiting the slowest rate of performance degradation among the compared methods. In the most constrained setting, with only 20 training samples per class, MSFF-Net attains an accuracy of 94.69%, outperforming all baseline models and demonstrating strong generalization capability in data-scarce environments.

A detailed analysis of the baseline methods further highlights the comparative advantages of MSFF-Net. Although the 1D-CNN-based VAF model benefits from architectural simplicity and low parameter count, it demonstrates limited accuracy in both standard and reduced-data conditions. This may be attributed to the use of large convolutional kernels, which restrict the model’s ability to capture fine-grained features. CDTFAFN performs competitively under full-data scenarios but undergoes more pronounced degradation in low-data settings, likely due to its reliance on complex signal transformations and multi-scale fusion structures. MSCNN maintains relatively stable performance with moderate data, yet under the most constrained condition, its accuracy falls by more than 11%p compared to MSFF-Net, indicating a lack of robustness when training data are severely limited.

In summary, MSFF-Net achieves a well-balanced trade-off among diagnostic accuracy, performance stability, and architectural simplicity. In comparison with other approaches, it requires a moderate number of learnable parameters and operates in the frequency domain using fast Fourier transform (FFT), which introduces temporal invariance and contributes to the stabilization of feature representations. These attributes enhance the model’s robustness under limited-data conditions and support its practical suitability for real-world fault diagnosis applications.

## 4. Conclusions

In this study, a lightweight diagnostic framework, termed MSFF-Net, was proposed for bearing fault classification based on frequency-domain features extracted from multiple sensors. By applying the fast Fourier transform (FFT) to both vibration and acoustic signals, the model extracts temporally invariant and spectrally stable representations. These representations are subsequently processed through a compact 1D convolutional neural network, enabling efficient feature learning with a relatively small number of parameters. Rather than employing complex architectural mechanisms such as multi-scale feature extraction or attention modules, a straightforward feature-level fusion strategy was adopted. Comparative experiments demonstrated that this approach outperformed single-modality configurations, validating the effectiveness of integrating complementary sensor information. Experimental results confirmed that MSFF-Net achieves high classification accuracy with low variability across trials, while maintaining a favorable trade-off among diagnostic performance, computational efficiency, and model complexity.

Feature-space visualizations and discriminability analyses further supported the model’s effectiveness, as the fused configuration exhibited improved class separability. Moreover, under small-sample conditions, MSFF-Net showed strong generalization capability, outperforming several representative baseline models and demonstrating robustness in data-scarce and data-limited conditions.

Despite these promising results, certain limitations remain. The current experiments were conducted using clean signals, without considering the effects of noise contamination. Additionally, the fusion strategy was limited to direct feature concatenation, without incorporating adaptive or learnable weighting mechanisms. Future research will explore the integration of attention-based or trainable fusion approaches and further assess the model’s robustness under noisy and variable operational environments.

## Figures and Tables

**Figure 1 sensors-25-04348-f001:**
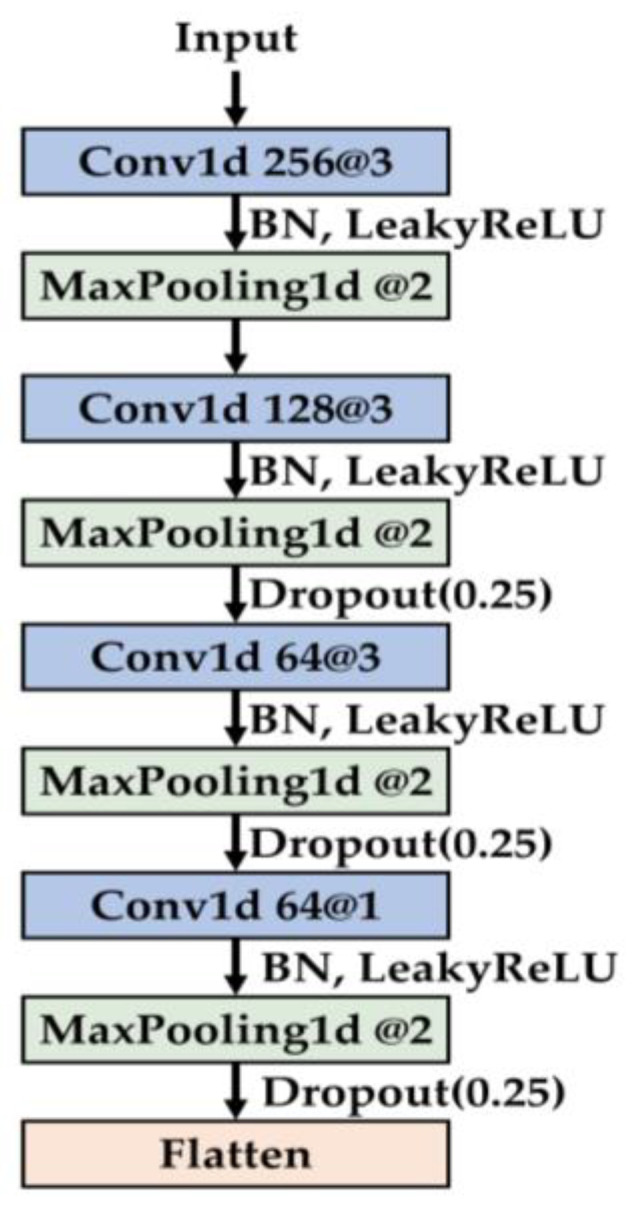
Architecture of the feature extraction module within MSFF-Net.

**Figure 2 sensors-25-04348-f002:**
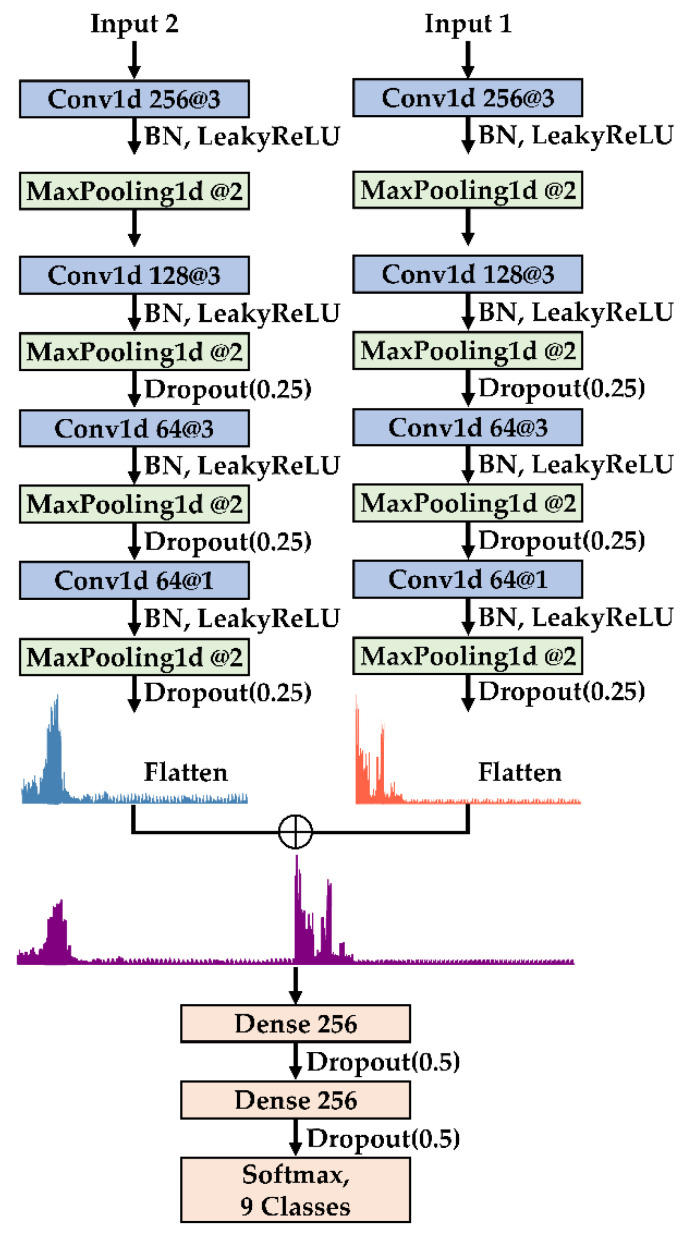
Overall architecture of MSFF-Net.

**Figure 3 sensors-25-04348-f003:**
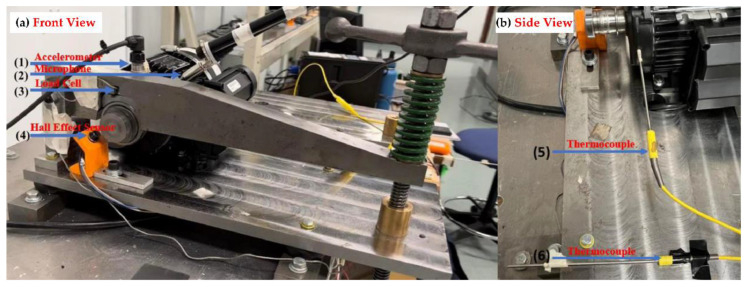
Schematic of the UORED-VAFCLS test rig used for vibration and acoustic signal acquisition [38].

**Figure 4 sensors-25-04348-f004:**
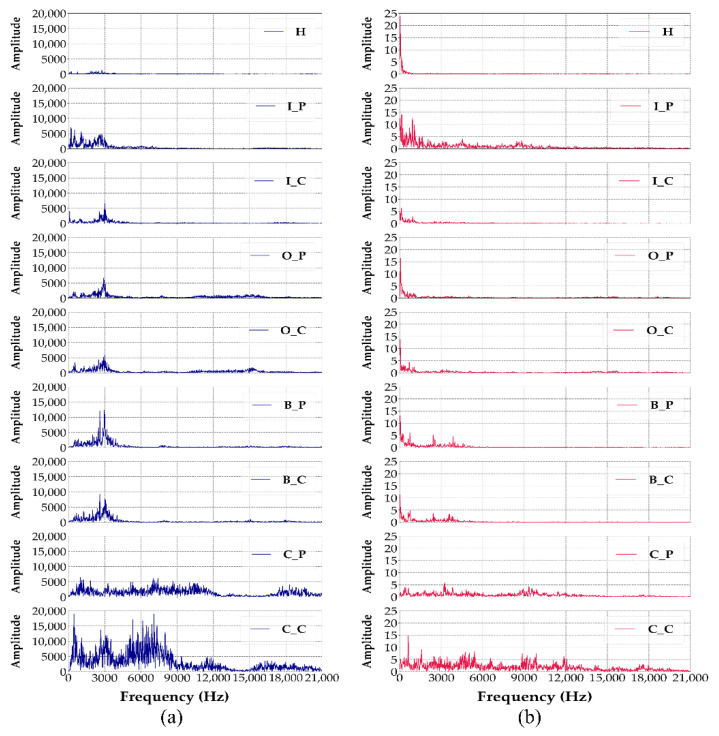
FFT power spectra of sensor signals in nine different bearing conditions: (**a**) vibration sensor, (**b**) acoustic sensor.

**Figure 5 sensors-25-04348-f005:**
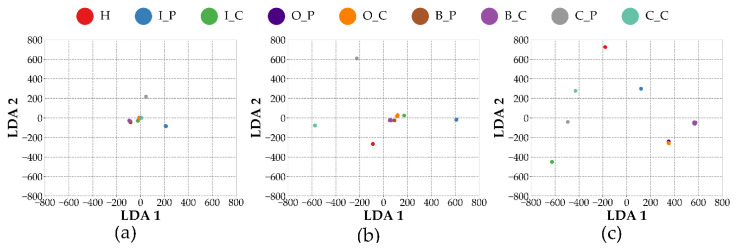
LDA projection of feature representations from different model configurations. (**a**) ACO, (**b**) VIB, and (**c**) VAF.

**Figure 6 sensors-25-04348-f006:**
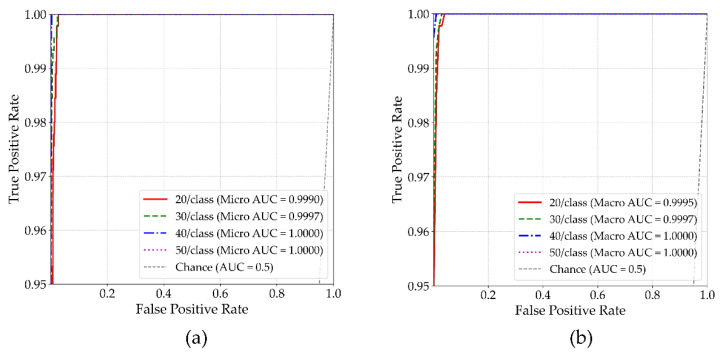
AUC results of the proposed MSFF-Net. (**a**) Micro-average AUC, (**b**) macro-average AUC.

**Figure 7 sensors-25-04348-f007:**
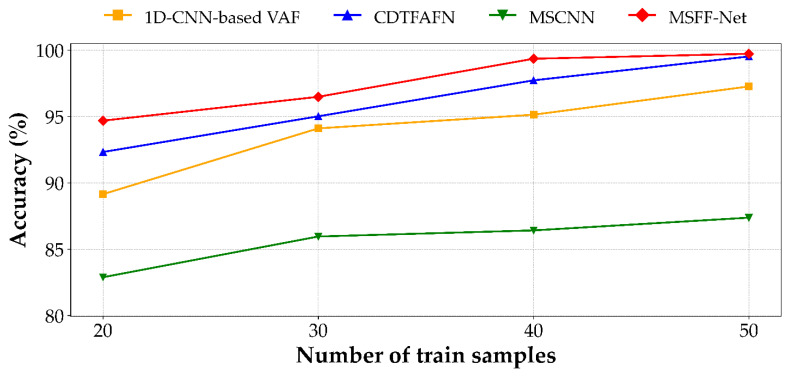
Performance of the proposed MSFF-Net and other comparison methods under small-sample conditions. The results for CDTFAFN are referenced from its original study [29] under the same experimental settings.

**Table 1 sensors-25-04348-t001:** Sample indices, training/testing sample counts, and label information used in the experiment.

Labels	Serial Number	Number of Training Samples	Number of Testing Samples
H	H_1_0	50	50
I_P	I_1_1	50	50
I_C	I_1_2	50	50
O_P	O_1_1	50	50
O_C	O_1_2	50	50
B_P	B_1_1	50	50
B_C	B_1_2	50	50
C_P	C_1_1	50	50
C_C	C_1_2	50	50

**Table 2 sensors-25-04348-t002:** Classification performance stability of the MSFF-Net model.

Metrics	Test Accuracy	Precision	Recall	Specificity	F1-Score
Mean	99.73%	99.74%	99.73%	99.97%	99.73%
Max.	100%	100%	100%	100%	100%
Min.	99.33%	99.34%	99.33%	99.92%	99.33%
Standard deviation	0.0025	0.0024	0.0025	0.0003	0.0025

**Table 3 sensors-25-04348-t003:** Comparing single-sensor models (VIB and ACO) with the proposed multi-sensor fusion model (VAF).

Method	Average Accuracy	Precision	Recall	Specificity	F1-Score
ACO	85.91%	85.83%	85.91%	98.24%	84.35%
VIB	98.40%	98.66%	98.40%	99.80%	98.36%
VAF	99.73%	99.74%	99.73%	99.97%	99.73%

**Table 4 sensors-25-04348-t004:** Quantitative discriminability metrics at final representation stage.

Processing Stage	Method	Intra-Class	Inter-Class	Inter/Intra
Final representation	ACO	7.9829	52.7540	6.6391
	VIB	5.4400	49.9306	8.0017
	VAF	6.6187	62.8282	9.4582

**Table 5 sensors-25-04348-t005:** Performance comparison of the proposed model and baseline methods. Results for CDTFAFN are adopted from [29], as it follows the same dataset partitioning.

Methods	Average Accuracy	Standard Deviation	Elapsed Time (s)	Number of Parameters (M)
1D-CNN-based VAF	97.27%	0.0131	30.07	1.7
CDTFAFN	99.53%	0.0727	71.03	3.4
MSCNN	87.38%	0.0417	70.02	0.3
MSFF-Net	99.73%	0.0025	49.69	2.4

**Table 6 sensors-25-04348-t006:** Average classification accuracy of different baseline methods under small-sample conditions. The results for CDTFAFN are as reported in [29].

Method	50 Samples/Class	40 Samples/Class	30 Samples/Class	20 Samples/Class
1D-CNN-based VAF	97.27%	95.14%	94.11%	89.15%
CDTFAFN	99.53%	97.73%	95.02%	92.33%
MSCNN	87.38%	86.42%	85.96%	82.89%
MSFF-Net	99.73%	99.36%	96.49%	94.69%

## Data Availability

Data will be made available on request.

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
