# Peer review of "MSFF-Net: Multi-Sensor Frequency-Domain Feature Fusion Network with Lightweight 1D CNN for Bearing Fault Diagnosis"

_sensors, 2025, doi:10.3390/s25144348_

Round 1
Reviewer 1 Report
Comments and Suggestions for Authors In this manuscript, the authors propose an Interpretable Frequency-Domain Multi-Sensor Fusion Network for Bearing Fault Diagnosis. Overall, the manuscript is well-structured, and the introduction is well-written. However, significant revisions are needed in the methodology and experimental sections. The specific comments are as follows:- The layout of the introduction is inappropriate. For example, the fourth paragraph consists of only one sentence, which is too short.
- It is good that the authors summarize the existing problems in the literature review. However, not all the problems identified by the authors are fully addressed in this study.
- The figures in the manuscript are too blurry, and readers cannot discern the details in the images.
- In the methodology section, the authors do not use any formulas to describe the method, which is unscientific.
- In the experimental section, what is the motivation for the authors to perform Fourier transforms on the audio data?
- In the experimental section, the authors provide too little introduction to the state-of-the-art methods. A more comprehensive introduction would facilitate a fairer comparison. Moreover, comparing with only one state-of-the-art method is far from sufficient.
- One of the core aspects of this manuscript is the interpretability of the diagnostic model. However, the analysis of interpretability in the experimental section is far from adequate. I suggest that the authors add more experimental result analyses to illustrate the interpretability of the model. Personally, I do not believe that this model can generate interpretability. Therefore, I suggest that the authors revise the title.
- Why does the proposed method have good generalization performance with small samples? I hope the authors can provide more analysis on this.
Author Response
Dear Reviewers,
We would like to express our sincere gratitude to the reviewers for their time, effort, and constructive feedback. Your insightful comments have been instrumental in enhancing the scientific clarity and overall quality of our manuscript. We have carefully addressed all the issues raised and revised the manuscript accordingly. Detailed responses to each comment, along with corresponding page and line references, are provided below. We deeply appreciate your valuable contributions to the advancement of this work.
Reviewer 1.
- The layout of the introduction is inappropriate. For example, the fourth paragraph consists of only one sentence, which is too short.
- We appreciate the reviewer’s thoughtful comment regarding the structural inconsistency in the Introduction. In response, we have integrated the original fourth paragraph previously a single sentence into a preceding paragraph to improve narrative cohesion. Additionally, we have expanded the discussion on the motivation and relevance of multi-sensor fusion, thereby enhancing both the logical flow and informational richness of the Introduction. These revisions can be found on Pages 2–3, Lines 65–122 of the revised manuscript.
- It is good that the authors summarize the existing problems in the literature review. However, not all the problems identified by the authors are fully addressed in this study.
- We appreciate the reviewer’s comment. Among the limitations identified in the literature, several were directly addressed in this study. Specifically, the structural complexity of existing methods and their dependence on domain-specific preprocessing were mitigated using a simplified 1D CNN architecture and FFT-based automatic feature extraction. In addition, the generalization capability under limited-data conditions was empirically validated through dedicated experiments.
- Nonetheless, certain challenges—such as the development of adaptive fusion strategies based on attention mechanisms and ensuring robustness in noisy environments—remain beyond the scope of the present study. In response to the reviewer’s suggestion, we have explicitly acknowledged these unresolved issues as limitations and proposed them as directions for future research in Section 4 Conclusions, Pages 17, Lines 631–636 of the revised manuscript.
- The figures in the manuscript are too blurry, and readers cannot discern the details in the images.
- We appreciate the reviewer’s comments regarding the low resolution of the figures, which may have hindered visual clarity. In response, all figures (Figures 1–7) have been regenerated at a resolution of 600 dpi or higher to ensure sharper image quality and more effective visual communication. Furthermore, the axes, legends, and annotations within each figure have been refined to meet publication standards
- In the methodology section, the authors do not use any formulas to describe the method, which is unscientific.
- We sincerely appreciate the reviewer’s comment regarding the absence of mathematical formulation in the methodology section. We fully agree that incorporating formal mathematical expressions is essential for enhancing the scientific rigor and reproducibility of the proposed approach. In response, we have added a series of key equations, numbered as Equations 3 through 8, throughout Section 2 titled Theory and Methodology, spanning Pages 5–6 and Lines 191–232 in the revised manuscript. These additions provide a more structured and precise explanation of the underlying framework.
- Specifically, we have included formulations for Z-score normalization, one-dimensional convolution, the Leaky ReLU activation function, max pooling, feature concatenation, fully connected layers, softmax, and cross-entropy loss. Each equation is accompanied by a concise interpretation and is embedded within the algorithmic flow to ensure clarity and completeness. For example, the convolution and activation processes are now described as follows:
- The remaining equations follow the same format, covering max pooling, concatenation, and fully connected operations with the ReLU activation function.
- These mathematical formulas were made to strengthen the theoretical foundations of our method and to make the algorithmic process more transparent, interpretable, and reproducible.
- In the experimental section, what is the motivation for the authors to perform Fourier transforms on the audio data?
- We sincerely thank the reviewer for raising this insightful question regarding the rationale behind applying the Fourier Transform to the acoustic signal. As extensively documented in the literature, faults in rotating machinery often generate periodic components and energy concentrations at characteristic fault frequencies, which are effectively captured in the frequency domain. These spectral signatures are not only prominent in vibration signals but are also present in airborne acoustic signals.
- Considering this, we applied the Fast Fourier Transform (FFT) to the time-domain acoustic signals in order to extract fault-relevant features in the frequency domain. This transformation offers several advantages, including invariance to time shifts and enhanced robustness to noise and transient fluctuations. Moreover, by applying the same spectral transformation to both vibration and acoustic signals, we aimed to reduce the modality gap between these heterogeneous sources, enabling more consistent and comparable feature representations for multi-sensor fusion.
- As detailed in Section 3.2, Performance of the Proposed Method, on Pages 10–11, Lines 346–369, although the acoustic signal alone yielded slightly lower classification accuracy than the vibration signal, their combination in the frequency domain led to substantial improvements in class separability. This result is further corroborated by the LDA-based feature analysis presented in Figure 5 and Table 4, which illustrate the enhanced discriminability across fault categories. We believe that this spectral-domain transformation contributes to greater training stability and improved generalization performance of the proposed model.
- In the experimental section, the authors provide too little introduction to the state-of-the-art methods. A more comprehensive introduction would facilitate a fairer comparison. Moreover, comparing with only one state-of-the-art method is far from sufficient.
- We thank the reviewer for this important suggestion concerning the need for a broader and more balanced comparison with state-of-the-art methods. In response, we have made substantial revisions to the manuscript to address this concern comprehensively.
- First, we expanded the discussion of existing diagnostic techniques based on multi-sensor fusion in Section 1, Introduction, spanning Pages 2–3 and Lines 65–122, to provide a more complete overview of prior work. This expanded literature review helps clarify the novelty and positioning of our proposed approach within the current research landscape.
- Second, in the experimental section (Pages 12, Lines 434–439), we have provided a detailed explanation of the state-of-the-art method, CDTFAN. Furthermore, our comparative analysis has been extended beyond CDTFAFN [29] to include a broader spectrum of representative models. This includes both single-sensor baselines—based on vibration-only and acoustic-only modalities—as well as multiple fusion-based approaches, enabling a more comprehensive performance evaluation. The details are as follows:
- In Section 3.2, Pages 10-11, Lines 346–369, we compare our model with single-modal baselines to assess the contribution of each sensor modality.
- In Section 3.3, Pages 12–14, Lines 417–495, we present comparative results against multiple state-of-the-art fusion methods under standard data conditions.
- In Section 3.4, Pages 14 –16, Lines 497–570, we evaluate model performance under small-sample training scenarios to validate robustness and generalizability.
- These additional revisions collectively ensure a fairer and more comprehensive evaluation of our proposed method relative to existing alternatives, thereby strengthening the validity of our experimental results.
- One of the core aspects of this manuscript is the interpretability of the diagnostic model. However, the analysis of interpretability in the experimental section is far from adequate. I suggest that the authors add more experimental result analyses to illustrate the interpretability of the model. Personally, I do not believe that this model can generate interpretability. Therefore, I suggest that the authors revise the title.
- We sincerely thank the reviewer for raising this important concern regarding the interpretability of the proposed model. In this manuscript, our use of the term “interpretability” refers to the ability to provide indirect insight into the model’s decision process by analyzing the separability of feature-level representations across fault classes. Specifically, we employed Linear Discriminant Analysis (LDA) to quantify how features extracted by the model contribute to enhanced class-wise discriminability, which, in our view, offers a limited but meaningful form of interpretability. This is distinct from more conventional definitions of interpretability involving attention visualization or direct decision-path tracing.
- However, we fully agree with the reviewer that this narrower interpretation does not meet the typical expectations of interpretability, namely, the ability to provide causal or human-understandable explanations for individual predictions. To avoid confusion and improve the clarity of the manuscript, we have revised the title and removed the term “Interpretable” accordingly. Additionally, we have clarified the intended scope of interpretability in Section 3.2, Performance of the Proposed Method, on Pages 11–12, Lines 370–415, and in Section 4, Conclusion, on Page 17, Lines 626–630. In these revisions, we emphasized that our analysis pertains to the separability of feature-level representations, rather than to model transparency in the conventional or causal sense.
- Why does the proposed method have good generalization performance with small samples? I hope the authors can provide more analysis on this.
- We thank the reviewer for this valuable comment. In response, we have expanded the discussion in Section 3.4, titled Small Sample Experiment, spanning Pages 14–16 and Lines 497–570 of the revised manuscript. This revision provides a more detailed analysis of the underlying factors contributing to the strong generalization performance of the proposed MSFF-Net model under limited-data conditions.
- There are three main factors contributing to this robustness:
- Lightweight architecture: Unlike complex deep models that require large amounts of training data to avoid overfitting, our model is based on a simplified 1D CNN structure with a small number of parameters and reduced computational cost. This design enables stable training even with small datasets.
- Sensor fusion in the frequency domain: The vibration and acoustic signals are transformed into the frequency domain and subsequently fused at the feature level. This approach enables the model to effectively leverage complementary information from the two modalities, thereby improving its ability to distinguish between normal and faulty conditions. This conclusion is substantiated by the LDA-based visualization results presented in Figure 5 and the quantitative discriminability metrics reported in Table 4, both of which demonstrate enhanced class separability under the fused configuration. Moreover, the frequency-domain representation provides time-invariant properties that contribute to faster convergence and more stable model training.
- Progressive feature compression: The MSFF-Net architecture progressively reduces both channel size and feature dimensionality, which helps eliminate redundant information and retain only the most salient features. This contributes to efficient representation learning, particularly beneficial in small-data regimes.
- Given that real-world industrial environments often lack sufficient labeled fault data, we believe the proposed architecture offers high practical value due to its ability to maintain accuracy with limited training samples.
- Response to Comments on the Quality of English Language. The English could be improved to more clearly express the research.
- According to the reviewer’s comments, we have cautiously revised the expression of the English in order to clearly express our research. As well, the English expressions in the revised manuscript were thoroughly revised using a professional academic proofreading service to meet the standards required for scholarly publications.

Reviewer 2 Report
Comments and Suggestions for Authors
Dear authors, thank you for well-structured and well-explained article on the topic multi-sensor fusion of bearing faults. Right of the bat I must say that I do not have any suggestions for improvement regarding the introduction and methodology. Simply fantastic in providing explainable and down-to-earth elaboration that can be found useful both for early career researchers, i.e., data scientists and professionals dealing with CNNs. Simply put, this is one of the best explanations of the architecture for feature selection and architecture of DL I've had privilege to review. One minor suggestions regarding the figures (although it may not be your own fault, maybe do to upload and transformation to pdf) to increase visibility by dpi or resolution since all figures in the paper appear blurry, at least from my side. So, I would argue to increase the visibility.
Also, regarding the results I do have some small suggestions and questions. I would recommend providing results of (AU)ROC over perofrmed metrics as these are particularly good in settings of class imbalance as it is commonly problem in identifying imbalanced failure count, as I'm sure you are pretty much aware, especially when the goal is threshold-independent assessment. Also, (maybe I've missed it somewhere in the paper) what activation function have you used? I see leaky relu for features, how about class faults?
I see that you've used LDA over traditionally reported UMAPs and t-SNEs, could you add more rationale for such? Have you checked how much does it offer better visualisation over other linear data visualisation (and reduction) techniques (e.g., PCA)? Make sure you use consistent font in figures as in text (i thinks it palatino).
I do not have any major suggestions for concluding remarks because authorsi ncluded limitations and conclusive findings. Perhaps maybe a small suggetions would be to add implications for practice and academia and contributions thereof. The paper is well-written.
Author Response
Dear Reviewers
We would like to express our sincere gratitude to the reviewers for their time, effort, and constructive feedback. Your insightful comments have been instrumental in enhancing the scientific clarity and overall quality of our manuscript. We have carefully addressed all the issues raised and revised the manuscript accordingly. Detailed responses to each comment, along with corresponding page and line references, are provided below. We deeply appreciate your valuable contributions to the advancement of this work.
Reviewer 2.
- Dear authors, thank you for well-structured and well-explained article on the topic multi-sensor fusion of bearing faults. Right of the bat I must say that I do not have any suggestions for improvement regarding the introduction and methodology. Simply fantastic in providing explainable and down-to-earth elaboration that can be found useful both for early career researchers, i.e., data scientists and professionals dealing with CNNs. Simply put, this is one of the best explanations of the architecture for feature selection and architecture of DL I've had privilege to review. One minor suggestions regarding the figures (although it may not be your own fault, maybe do to upload and transformation to pdf) to increase visibility by dpi or resolution since all figures in the paper appear blurry, at least from my side. So, I would argue to increase the visibility.
- We appreciate the reviewer’s comments regarding the low resolution of the figures, which may have hindered visual clarity. In response, all figures (Figures 1–7) have been regenerated at a resolution of 600 dpi or higher to ensure sharper image quality and more effective visual communication. Furthermore, the axes, legends, and annotations within each figure have been refined to meet publication standards
- Also, regarding the results I do have some small suggestions and questions. I would recommend providing results of (AU)ROC over performed metrics as these are particularly good in settings of class imbalance as it is commonly problem in identifying imbalanced failure count, as I'm sure you are pretty much aware, especially when the goal is threshold-independent assessment. Also, (maybe I've missed it somewhere in the paper) what activation function have you used? I see leaky relu for features, how about class faults?
- Regarding the first point, although the datasets used in our study are relatively balanced in terms of class distribution, we agree that class imbalance can still arise implicitly in small-sample scenarios due to probabilistic sampling effects. This may lead to biased decision boundaries or underfitting in early training stages. In response to your suggestion, we have incorporated additional evaluation using the Area Under the ROC Curve (AUC) metric.
- As described in Section 3.4, Small Sample Experiment, on Pages 14–15 and Lines 505–533 of the revised manuscript, we progressively reduced the number of training samples per class to 50, 40, 30, and 20. For each setting, we computed both micro- and macro-average AUC scores to evaluate model performance under varying data scarcity levels. The corresponding results are now visualized in Figure 6 to provide a clear comparative overview. The results show that the model maintains consistently high classification reliability across varying training sample sizes, as reflected in both micro- and macro-averaged ROC curves. Although minor reductions in AUC occur at 30 and 20 samples per class, all values remain above 0.9990, indicating minimal impact. Given the near-identical shapes of the two curves, both global recognition ability and class-level uniformity are effectively preserved.
- Regarding the second question, we would like to clarify that the Leaky ReLU activation function is applied after each convolutional layer to support intermediate feature extraction, while the final classification output layer employs the softmax function to enable multi-class probability estimation. The definitions and roles of these activation functions have now been explicitly described using mathematical expressions in Section 2.3, One-Dimensional Convolutional Neural Network, on Page 5-6 and Lines 200–226 of the revised manuscript. For example, the Leaky ReLU function applied to each convolutional output is defined as follows:
- I see that you've used LDA over traditionally reported UMAPs and t-SNEs, could you add more rationale for such? Have you checked how much does it offer better visualisation over other linear data visualisation (and reduction) techniques (e.g., PCA)? Make sure you use consistent font in figures as in text (i thinks it palatino).
- We appreciate the reviewer’s insightful comment regarding the use of LDA in place of more commonly used dimensionality reduction techniques such as UMAP and t-SNE. We have revised Section 3.2, Performance of the Proposed Method, on Page 10-11 and Lines 370 to 385 of the revised manuscript to clarify the rationale behind this design choice.
- The primary purpose of applying Linear Discriminant Analysis (LDA) in our study was not merely for visual interpretation, but to quantitatively evaluate class separability within the high-dimensional feature space across different processing stages (input, intermediate, and output). Unlike UMAP or t-SNE—which are powerful for visualizing non-linear embeddings—LDA is explicitly designed to maximize the ratio of inter-class variance to intra-class variance. This property makes it particularly suitable for analyzing discriminability, a key factor in fault diagnosis.
- In the revised version of the manuscript, we have chosen to retain only the LDA visualization corresponding to the final output stage. This decision was made to improve focus and readability, as the final-stage feature space alone sufficiently reflects the enhanced separability achieved by the proposed model, as discussed on Pages 11–12, Lines 386–406 of the revised manuscript. While the earlier version included multi-stage analysis, we found that concentrating on the output layer still effectively supports the interpretability claims without introducing redundancy.
- Moreover, while PCA provides a linear projection preserving overall variance, it is unsupervised and does not take class labels into account, thereby limiting its capacity to assess class-level separation. In contrast, LDA leverages label information and yields interpretable projections optimized for class separation. In this sense, our use of LDA serves a statistical and diagnostic purpose, rather than only offering a visual heuristic.
- Lastly, we appreciate the reviewer’s observation regarding font consistency in the figures. In response, we have carefully revised all figure annotations to align with the body text font, Palatino, and ensured visual consistency across all figures in the revised version of the manuscript.
- I do not have any major suggestions for concluding remarks because the authors included limitations and conclusive findings. Perhaps maybe a small suggestion would be to add implications for practice and academia and contributions thereof. The paper is well-written.
- We sincerely thank the reviewer for the positive evaluation of our manuscript and for the helpful suggestion to enhance the concluding remarks. In accordance with the reviewer’s suggestion, we have revised Section 4, Conclusion, on Page 17, Lines 631 –637, to explicitly highlight both the practical and academic implications of this work.
- We emphasized that the proposed MSFF-Net architecture—due to its lightweight design and robust performance under small-data conditions—holds strong potential for real-world deployment in industrial fault diagnosis, where labeled data are often scarce. Furthermore, our feature-level fusion approach and its LDA-based quantitative analysis of class separability may provide a useful reference for the academic community in evaluating and developing future multi-sensor diagnostic models with interpretability-oriented performance assessments.

Reviewer 3 Report
Comments and Suggestions for Authors
The paper proposes a method for detecting bearing faults based on vibration and acoustic signals processed using the Short-Time Fourier Transform (STFT) and a 1D Convolutional Neural Network (1D-CNN). The manuscript is well-structured, providing a clear explanation of the proposed method.
However, there are minor issues to be addressed:
1) All Figures need a quality improvement; most of them are blurry and do not provide clear labels and legends.
2) Figures 1 and 2 should be rotated to a vertical layout.
3) Figure 4, insert the Y-axis label.
4) Figure 5 requires a deeper explanation; what is its utility in the proposed method?
5) The main problem in the article is to justify the minimal improvement against the SOTA technique. Since the article's goal is not to reduce the computational load, which is reduced by 30%.
6) How does the proposed method address the issue of interpretability? It is not completely clear.
Author Response
Dear Reviewers
We would like to express our sincere appreciation to the reviewers for their time, effort, and constructive feedback. Your insightful comments were invaluable in improving the scientific clarity and overall quality of our work. We carefully addressed all concerns raised, and have revised the manuscript accordingly, with detailed responses and page/line indications. Thank you once again for your meaningful contribution to the advancement of this research.
Reviewer 3.
- All Figures need quality improvement; most of them are blurry and do not provide clear labels and legends.
- We appreciate the reviewer’s comments regarding the low resolution of the figures, which may have hindered visual clarity. In response, all figures (Figures 1–7) have been regenerated at a resolution of 600 dpi or higher to ensure sharper image quality and more effective visual communication. Furthermore, the axes, legends, and annotations within each figure have been refined to meet publication standards
- Figures 1 and 2 should be rotated to a vertical layout.
- We have revised Figure 1 and Figure 2 by rotating them into a vertical layout, as recommended. We believe that the revised layout offers a clearer presentation of the model structure and processing pipeline.
Figure 1. Architecture of the feature extraction module within MSFF-Net.
Figure 2. Overall architecture of MSFF-Net.
- Figure 4, insert the Y-axis label.
- We thank the reviewer for pointing out the missing Y-axis label in Figure 4. We agree that the lack of a label may cause confusion. In response, we have explicitly added the Y-axis label “Amplitude” to each graph in Figure 4 to ensure clarity and improve interpretability.
Figure 4. FFT power spectra of sensor signals in 9 different bearing conditions: (a) vibration sensor; (b) acoustic sensor.
- Figure 5 requires a deeper explanation; what is its utility in the proposed method?
- We thank the reviewer for the helpful comment. In response, we have clarified the role and interpretive significance of Figure 5 in Section 3.2, Performance of the Proposed Method, on Pages 11-12, and Lines 390-406 of the revised manuscript.
- The purpose of Figure 5 is not only to serve as a general low-dimensional visualization of high-dimensional data, but also to provide quantitative insight into the discriminative capacity of the feature representations learned by the proposed MSFF-Net. By applying LDA to the feature-level outputs of both the proposed method and baseline single-sensor models, we assessed how well the learned features separate different fault classes in the projected space. This figure supports our claim that multi-sensor fusion at the feature level significantly enhances class separability, which is a core strength of the proposed architecture.
- The main problem in the article is to justify the minimal improvement against the SOTA technique. Since the article's goal is not to reduce the computational load, which is reduced by 30%.
- We appreciate the reviewer’s comment regarding the limited performance improvement over the SOTA method. The primary objective of this study is to propose an efficient fault diagnosis framework that does not rely on complex feature extraction or fusion mechanisms, while still achieving competitive performance.
- In the revised manuscript, we have included additional comparison experiments with several representative feature-level fusion methods beyond the original SOTA baseline. Notably, under small-sample conditions, as presented on Pages 15–16 and Lines 534–570 of the revised manuscript, the proposed model demonstrates greater stability and stronger generalization capability compared to both the state-of-the-art baseline and other competing methods. Given that collecting sufficient fault data in real industrial settings is often difficult, we believe that stable generalization under data-scarce scenarios constitutes an important and meaningful contribution.
- How does the proposed method address the issue of interpretability? It is not completely clear.
- We appreciate the reviewer’s comment regarding the clarity of the interpretability aspect. In our original manuscript, the term “interpretability” was used to refer to the ability to provide indirect insights into class-wise feature representations via Linear Discriminant Analysis (LDA). Specifically, the goal was to evaluate whether and how the learned features improve class separability, which in turn helps explain why the proposed model outperforms single-sensor baselines.
- Unlike attention-based or decision-path-level interpretability approaches, our focus was on quantitatively analyzing the discriminability of feature-level distributions to shed light on the model’s internal behavior. However, as the reviewer correctly pointed out, the conventional understanding of interpretability typically involves explaining why specific predictions are made and which input factors contribute to those decisions. We fully acknowledge that our approach does not meet this stricter definition.
- Another reviewer also raised a similar concern regarding the appropriateness of the term "interpretable." In response, we have removed the term from the manuscript title and revised the relevant expressions throughout the manuscript to more accurately reflect the nature of our approach. To avoid potential confusion and enhance clarity, we have explicitly clarified the scope and limitations of our use of the term interpretability in Section 3.2, Performance of the Proposed Method, on Page 11 and Lines 391–396, as well as in Section 4, Conclusion, on Page 17 and Lines 626–630. These revisions are intended to help readers more accurately understand how the concept of interpretability is defined and applied with

Round 2
Reviewer 1 Report
Comments and Suggestions for Authors
ACCEPT
Reviewer 2 Report
Comments and Suggestions for Authors
Dear authors thank you for considering my suggestions. I do not have any more suggestions.